# Protracted Tonsillitis as an Atypical Initial Manifestation of Methotrexate-Induced EBV-Positive Lymphoproliferative Disorder in Rheumatoid Arthritis: A Case Report and Literature Review

**DOI:** 10.3390/diagnostics15121517

**Published:** 2025-06-14

**Authors:** Ting-Shen Lin, Tang-Yi Tsao, Shih-Wei Chen, Min-Cheng Ko, Stella Chin-Shaw Tsai

**Affiliations:** 1Department of Otolaryngology, Tungs’ Taichung MetroHarbor Hospital, Taichung 435, Taiwan; t14412@ms3.sltung.com.tw (T.-S.L.); t7819@ms3.sltung.com.tw (S.-W.C.); 2Department of Pathology, Tungs’ Taichung MetroHarbor Hospital, Taichung 435, Taiwan; t3041@ms3.sltung.com.tw; 3Superintendent Office, Tungs’ Taichung MetroHarbor Hospital, Taichung 435, Taiwan; 4Department of Post-Baccalaureate Medicine, National Chung Hsing University, Taichung 402, Taiwan

**Keywords:** methotrexate, lymphoproliferative disorder, tonsillitis, rheumatoid arthritis, immunosuppression

## Abstract

**Background and Clinical Significance:** Methotrexate is widely used as a disease-modifying antirheumatic drug for rheumatoid arthritis (RA), yet prolonged immunosuppression may lead to rare complications, including Epstein–Barr virus (EBV)-positive lymphoproliferative disorders (LPDs). **Case Presentation:** We present the case of a 70-year-old woman with RA on chronic immunosuppressive therapy who developed symptoms resembling recurrent tonsillitis. CT imaging revealed bilateral necrotic palatine tonsils and extensive necrotic lymphadenopathy involving the cervical, mediastinal, and axillary regions. Bilateral tonsillectomy was performed due to concerns about malignancy or infection, and histopathology confirmed a polymorphic EBV-positive LPD with Hodgkin-like features, consistent with iatrogenic immunodeficiency-associated LPD. Methotrexate was subsequently discontinued, and the patient was managed conservatively without systemic chemotherapy. Clinical recovery was observed during follow-up. **Conclusions:** This case highlights the importance of considering methotrexate-associated LPDs in the differential diagnosis of atypical tonsillar infections in immunosuppressed patients, particularly when necrotic features or systemic lymphadenopathy are present. The pathogenesis may involve EBV reactivation under impaired immune surveillance due to methotrexate, leading to abnormal B-cell proliferation and clonal expansion. This case is contextualized through a comparative analysis of published reports, highlighting clinical features and treatment responses of methotrexate-associated EBV-positive LPDs in the form of a focused literature review.

## 1. Introduction

Methotrexate (MTX), a folate antagonist, is widely recognized as a first-line disease-modifying antirheumatic drug (DMARD) for rheumatoid arthritis (RA), owing to its anti-inflammatory efficacy, cost-effectiveness, and relatively favorable safety profile. In addition to RA, MTX is commonly used in other autoimmune and inflammatory conditions, such as psoriasis, systemic lupus erythematosus, and vasculitis [1]. Despite its widespread use and low-dose regimen in rheumatology, prolonged MTX therapy carries an established risk of rare but potentially serious adverse events, including hepatotoxicity, bone marrow suppression, pulmonary toxicity, and, more recently recognized, lymphoproliferative disorders (LPDs) [1,2].

Among these, iatrogenic immunodeficiency-associated lymphoproliferative disorders (IA-LPDs) have emerged as a distinct entity, predominantly affecting patients undergoing immunosuppressive therapy. IA-LPDs encompass a heterogeneous group of lymphoid proliferations that arise in the context of pharmacologic immune suppression and often regress upon discontinuation of the inciting agent. A substantial proportion of these disorders are associated with reactivation of latent Epstein–Barr virus (EBV), a ubiquitous herpesvirus known for its oncogenic potential in immunocompromised hosts [2,3,4]. EBV-driven LPDs in MTX-treated individuals exhibit diverse clinicopathologic features, ranging from EBV-positive mucocutaneous ulcers to polymorphic or monomorphic B-cell proliferations with or without Hodgkin-like morphology. These lesions often mimic acute infections or malignancies, complicating timely diagnosis. Extranodal involvement is common, with the oral cavity, gastrointestinal tract, skin, and lymphoid tissues being frequently affected sites [2,3,4,5]. Notably, the palatine tonsils, although rarely the initial site of presentation, can serve as a portal for EBV-induced lymphoid proliferation in immunosuppressed individuals.

The diagnostic challenge lies in the clinical overlap between benign infectious conditions and early-stage LPDs. In such cases, radiologic findings, histopathology, and EBV status by in situ hybridization are essential for diagnostic clarification. Moreover, increasing recognition of spontaneous regression of these disorders upon MTX withdrawal has shifted management paradigms toward a more conservative, immune-restorative approach in selected patients.

This case contributes to the expanding clinical spectrum of methotrexate-associated EBV-positive lymphoproliferative disorders by illustrating an uncommon initial manifestation involving the palatine tonsils. It highlights the diagnostic complexity in rheumatoid arthritis patients presenting with persistent or unusual oropharyngeal symptoms in the context of immunosuppression.

## 2. Case Report

A 70-year-old female with a longstanding history of seropositive RA presented with recurrent episodes of sore throat for two months. Her medications included celecoxib, methotrexate sodium, hydroxychloroquine sulfate, and sulfasalazine. A physical examination in the outpatient clinic revealed erythematous swelling with purulent exudate over the bilateral palatine tonsils. Laboratory evaluation revealed a white blood cell count of 7000/μL, which falls within the normal reference range (approximately 4000–10,000/μL). The erythrocyte sedimentation rate was elevated at 30 mm/h (normal: 0–20 mm/h), and the C-reactive protein level was also increased at 14.95 mg/L (normal: <5 mg/L), indicating the presence of systemic inflammation. Contrast-enhanced computed tomography (CT) of the neck and chest revealed bilateral palatine tonsils with necrosis (Figure 1). Additionally, multiple necrotic lymphadenopathies were identified in the cervical region, superior mediastinum, and both axillary areas. Due to concern for underlying malignancy or infectious complications, the patient underwent bilateral tonsillectomy. Histopathological examination revealed features consistent with an EBV-positive lymphoproliferative disorder, polymorphic Hodgkin-like type, compatible with the diagnosis of an iatrogenic immunodeficiency-associated LPD (Figure 2). However, the tonsillar wound persisted with ulceration and necrosis more than one month after surgery, prompting further evaluation and clinical concern. Methotrexate was discontinued and the patient was managed conservatively with close monitoring. Subsequent follow-up showed clinical improvement, and no systemic chemotherapy was initiated (Figure 3).

## 3. Discussion

Iatrogenic immunodeficiency-associated lymphoproliferative disorders (IALPDs) are recognized as rare complications of long-term immunosuppression, particularly in patients treated with methotrexate [1]. EBV is frequently implicated in these cases and may drive abnormal B-cell proliferation in immunocompromised hosts [4,5]. The initial presentation of this patient mimicked common tonsillitis with erythematous, pus-coated palatine tonsils. While tonsillitis is a frequent diagnosis in both pediatric and adult populations, most cases are viral or bacterial, with a favorable response to supportive or antibiotic treatment [6]. Certain atypical features may indicate alternative diagnoses. Findings such as asymmetric tonsillar necrosis, persistent symptoms, and associated regional or generalized lymphadenopathy should raise clinical suspicion for lymphoproliferative disorders [4]. In this case, the presence of necrotic tonsils along with widespread lymphadenopathy further increased the concern for a possible malignant process.

Although lymphoid tissues such as tonsils and lymph nodes are common sites, MTX-associated LPDs may also arise in uncommon locations. Yamaguchi et al. reported the first known case of MTX-LPD in the sacrum, demonstrating that immunosuppression-related lymphoproliferation can present in unusual anatomic sites [7]. EBV-positive mucocutaneous ulcer (EBV-MCU), a localized lymphoproliferative disorder associated with immunosuppressive therapy, shares histologic and clinical features with our case. In a landmark study of 26 cases, Dojcinov et al. characterized EBV-MCU as a distinct clinicopathologic entity arising in the setting of various forms of immunosuppression, including methotrexate [2]. However, MTX-associated B-cell LPDs are not limited to EBV positive forms. Satou et al. reported on primary cutaneous MTX-associated B-cell LPDs without EBV involvement, emphasizing the pathological and immunophenotypic heterogeneity of these disorders [5]. More recently, Shiraiwa et al. confirmed that EBV-positive mucosal lesions often present with favorable clinical outcomes and may regress spontaneously following MTX cessation [4]. In addition, studies have shown variability in EBV association in classical Hodgkin lymphoma (cHL) at different disease stages, raising questions about the role of EBV in pathogenesis and recurrence patterns [8]. Delecluse et al. reported the complete disappearance of EBV in the relapse phase of cHL in a previously EBV-positive patient, suggesting that EBV presence may fluctuate throughout disease progression [9]. Notably, a subset of MTX-associated LPDs may regress spontaneously upon cessation of the immunosuppressive agent, obviating the need for chemotherapy. Salloum et al. reported multiple cases of spontaneous regression of LPDs in patients treated with methotrexate, reinforcing the potential reversibility of these disorders upon drug withdrawal [10].

A comparison of published cases of MTX-LPDs is summarized in Table 1. Across diverse anatomical sites, including oral mucosa, skin, bone, and lymphoid tissues, EBV has emerged as a recurrent etiologic factor, particularly in patients undergoing chronic immunosuppressive therapy for autoimmune diseases such as rheumatoid arthritis. In cases where EBV positivity was confirmed, spontaneous regression following methotrexate withdrawal was frequently observed, reducing the need for systemic chemotherapy. Histologically, most reported cases demonstrated polymorphic features, often with Hodgkin-like morphology. EBV-positive mucocutaneous ulcers and polymorphic B-cell LPDs are among the most common subtypes reported in this setting. Notably, EBV-negative MTX-LPDs, while less frequent, tend to display greater heterogeneity in pathology and may require additional intervention such as chemotherapy [5]. These findings highlighted two important clinical implications. First, EBV testing by Epstein–Barr encoding RNA (EBER) in situ hybridization should be routinely performed in suspected MTX-LPD cases to aid diagnosis and prognostication. Second, withdrawal of methotrexate alone is often sufficient to induce regression, particularly in EBV-positive cases, reinforcing the value of early recognition and drug cessation. The present case is unique in that bilateral necrotic tonsillitis served as the initial manifestation of MTX-LPD, an extremely rare presentation. Its spontaneous resolution following MTX withdrawal aligns with patterns seen in other EBV-driven LPDs and further expands the clinical spectrum of this condition.

### 3.1. Mechanisms of EBV Reactivation and Immune Escape

Epstein–Barr virus (EBV), a highly endemic gamma-herpesvirus, establishes lifelong latency in memory B cells following primary infection [11]. In immunocompetent individuals, EBV is typically controlled by cytotoxic CD8+ T-cell surveillance, maintaining the virus in a latent, non-pathogenic state. However, in immunocompromised conditions, particularly iatrogenic immunosuppression from MTX, the host’s immune surveillance is impaired, allowing reactivation and proliferation of EBV-infected B cells, which can lead to LPD [12,13,14]. Methotrexate has been shown to exert immunosuppressive effects not only through inhibition of folate metabolism but also by suppressing T-cell activation and proliferation [15]. Specifically, MTX impairs CD8+ T-cell-mediated cytotoxicity and reduces interleukin-2 production, weakening control over EBV-infected B-cell clones. Additionally, MTX may alter the local microenvironment by increasing regulatory T-cell activity, which further suppresses effective antiviral immune responses.

EBV possesses sophisticated immune evasion strategies that enhance its survival during reactivation [13]. For instance, latent EBV proteins such as LMP1 and LMP2A mimic cellular survival signals, promoting B-cell proliferation and resistance to apoptosis. Moreover, EBV can downregulate MHC class I and class II molecules on infected B cells, thereby reducing their visibility to cytotoxic T lymphocytes and helper T cells. Some EBV strains may also induce production of IL-10, a cytokine with known immunosuppressive properties, further dampening host immune responses [16,17]. The interplay between EBV and an immunosuppressed host creates a permissive environment for clonal B-cell expansion. In cases such as this, where the histology reveals polymorphic LPD with Hodgkin-like features, it is plausible that EBV-driven activation of oncogenic pathways, including NF-κB and JAK/STAT signaling, contributes to the proliferation and survival of malignant or pre-malignant B-cell clones [16,17].

### 3.2. Epstein–Barr Virus and Rheumatoid Arthritis

The interplay between EBV and RA has been extensively investigated, with growing evidence supporting a possible pathogenic link. EBV infection is nearly universal, but individuals with RA appear to exhibit distinct immunologic responses to the virus. Multiple studies have demonstrated that patients with RA show elevated EBV viral loads, heightened EBV-specific antibody titers, and impaired cytotoxic T-cell responses against EBV-infected B cells when compared to healthy controls [12,13]. One proposed mechanism suggests that chronic immune activation in RA creates a permissive environment for latent EBV reactivation, particularly in the context of immunosuppressive therapies such as MTX. RA patients often exhibit defective CD8+ T-cell-mediated immunity against EBV, which may lead to the survival of infected B cells and accumulation of viral antigens within synovial tissues [15]. This persistence could promote chronic antigenic stimulation, thereby contributing to the maintenance and amplification of autoimmune processes.

Molecular mimicry has also been implicated. EBV-encoded proteins share structural similarities with host autoantigens, potentially leading to cross-reactive immune responses that target self-tissues. For instance, EBV nuclear antigen 1 (EBNA1) contains peptide motifs with homology to citrullinated peptides such as filaggrin and type II collagen, which are key targets in RA-associated autoimmunity [12,13]. Additionally, EBV’s ability to infect and immortalize B lymphocytes plays a critical role in RA pathogenesis. These EBV-infected B cells may act as reservoirs of autoreactivity, participating in autoantigen presentation, cytokine release, and autoantibody production. In genetically predisposed individuals, particularly those carrying HLA-DRB1 shared epitope alleles, the immunologic response to EBV is believed to be skewed toward a pathogenic, pro-inflammatory phenotype, increasing the likelihood of both RA onset and EBV-driven lymphoproliferative transformation [13,14].

In the case presented here, the convergence of chronic RA, latent EBV infection, and prolonged methotrexate-induced immunosuppression likely resulted in a synergistic breakdown of immune regulation. This culminated in the clinical emergence of an EBV-positive lymphoproliferative disorder with tonsillar involvement. Understanding the multifaceted relationship between EBV and RA may help identify high-risk populations and inform tailored monitoring strategies for patients undergoing immunosuppressive treatment.

### 3.3. MTX Cessation and LPD Regression

A defining and clinically consequential feature of MTX-LPDs is their potential for spontaneous regression following the withdrawal of immunosuppressive therapy. This phenomenon, which is particularly prominent in EBV-positive cases, has been well documented across multiple series and case reports, reinforcing the concept of MTX-LPD as a reversible immunologic dysregulation rather than a fixed neoplastic process (Table 1).

Salloum et al. [10] first described multiple instances of complete remission in patients with RA-associated LPDs following MTX cessation, without the need for cytotoxic chemotherapy. Subsequent studies, including the cohort by Dojcinov et al. [2], further validated this observation in EBV-positive mucocutaneous ulcers and polymorphic LPDs, where the majority of patients demonstrated either partial or complete regression with immunosuppressant withdrawal alone. The consistency of these findings has since led to the recognition of MTX cessation as both a diagnostic and therapeutic maneuver in the management of iatrogenic immunodeficiency-associated LPDs. Mechanistically, regression is attributed to the restoration of immune competence, specifically the reconstitution of cytotoxic CD8+ T-cell activity against EBV-infected B cells. Saito et al. [15] provided immunologic evidence demonstrating that recovery of Th1 and CD8+ lymphocyte subsets coincided temporally with LPD resolution following MTX discontinuation. This suggests that MTX impairs immune surveillance sufficiently to permit unchecked B-cell proliferation in the presence of latent EBV, a process that can reverse upon removal of the immunosuppressive agent. Clinically, the response to MTX withdrawal may serve as a practical means of risk stratification [12,18]. Cases demonstrating regression often obviate the need for systemic chemotherapy, thereby sparing patients from overtreatment. However, this approach necessitates vigilant follow-up, as not all cases regress, particularly those with EBV-negative histology, aggressive monomorphic features, or systemic involvement. For such cases, prompt escalation to oncologic therapies may still be warranted.

In the present case, the complete clinical resolution observed following methotrexate cessation without any adjunctive therapy is consistent with prior literature and underscores the importance of early drug withdrawal in suspected MTX-LPD. This therapeutic strategy is particularly relevant for EBV-positive, polymorphic cases presenting in an indolent fashion, where a conservative, immune-restorative approach may be not only sufficient, but preferable. Spontaneous regression following withdrawal of MTX, as seen in our patient, supports the hypothesis that immune restoration is sufficient to re-establish viral control and induce apoptosis of EBV-infected cells. This phenomenon has been well documented in EBV-positive mucocutaneous ulcers and other IA-LPD subtypes, reinforcing the notion that immunosuppression, not genetic mutation alone, is a driving factor in the pathogenesis of these lesions [2,3,4].

### 3.4. Diagnostic Alternatives to Tonsillectomy

While tonsillectomy provided both symptomatic relief and diagnostic tissue in this case, the presence of necrotic cervical lymphadenopathy on imaging raises the question of whether a core needle biopsy (CNB) or excisional lymph node biopsy could have been considered initially. These less invasive approaches are widely used in the evaluation of suspected lymphoproliferative disorders, including EBV-positive cases, and can yield sufficient material for histopathologic and molecular analysis [19].

In MTX-LPDs, especially those involving peripheral nodes, image-guided CNB is often preferred when lesions are accessible and systemic symptoms are absent. However, sampling error remains a limitation, particularly in morphologically heterogeneous diseases. In our case, the tonsils were the dominant symptomatic site and likely represented the primary disease focus, justifying the decision to perform a tonsillectomy. Nonetheless, this case highlights the importance of tailoring diagnostic strategies based on accessibility, clinical urgency, and expected diagnostic yield.

### 3.5. Statement of Impact and Relevance

This case report highlights the clinical significance of protracted tonsillitis as an atypical initial presentation of methotrexate-induced EBV-positive LPD in a patient with rheumatoid arthritis. The presence of necrotic tonsillar lesions and widespread lymphadenopathy initially mimicked a routine infectious process, delaying recognition of the underlying pathology. Histological confirmation and complete clinical recovery following methotrexate withdrawal emphasize the importance of early diagnostic consideration of LPD in immunosuppressed patients. This case contributes to the current understanding of iatrogenic immunodeficiency-associated LPDs and demonstrates that early drug cessation can result in disease regression, potentially avoiding unnecessary chemotherapy.

### 3.6. Limitations and Future Direction

While this case report illustrates a reversible EBV-positive LPD following methotrexate withdrawal, several limitations exist. EBV viral load quantification was not performed, and clonality studies could have further clarified disease classification. Additionally, long-term follow-up is ongoing and recurrence risk remains unclear. Future studies should explore predictive biomarkers to identify the RA patients at highest risk of LPD, and define monitoring protocols for early detection.

## 4. Conclusions

The presented case emphasizes the need for clinical vigilance when evaluating patients with rheumatoid arthritis on long-term methotrexate therapy who develop atypical tonsillar symptoms and necrotic lymphadenopathy. While invasive procedures such as tonsillectomy may not be routinely necessary, tissue biopsy can provide diagnostic clarity in selected cases. Early recognition and timely withdrawal of methotrexate are crucial steps in managing methotrexate-associated EBV-positive LPDs.

## Figures and Tables

**Figure 1 diagnostics-15-01517-f001:**
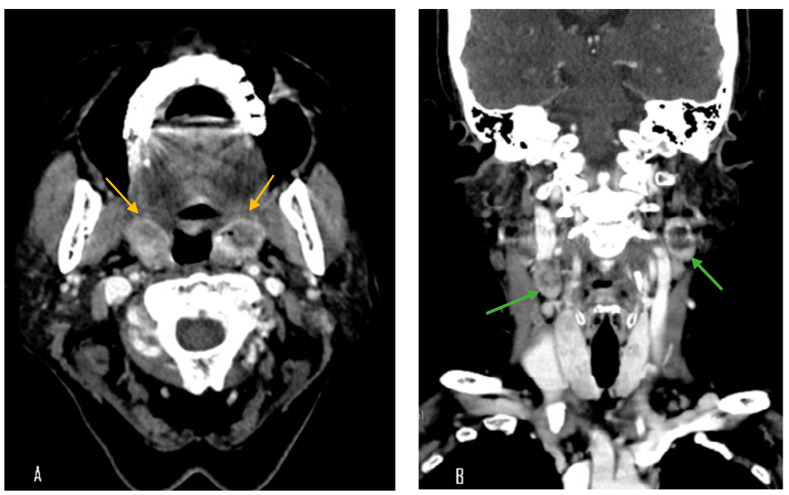
Contrast-enhanced computed tomography (CT) images. (**A**) Axial view showing bilateral palatine tonsillar necrosis (yellow arrow). (**B**) Coronal view revealing multiple necrotic lymphadenopathies in the cervical region (green arrow).

**Figure 2 diagnostics-15-01517-f002:**
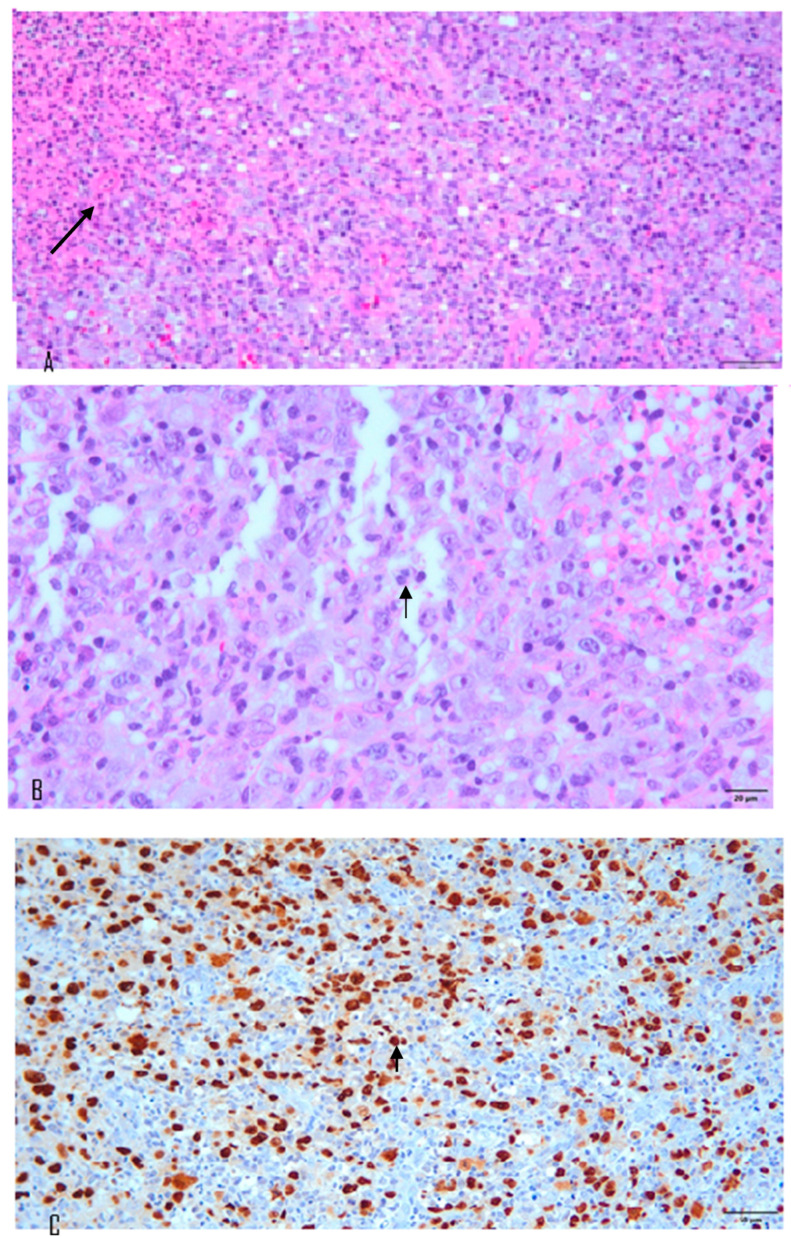
(**A**) Histologic features include architectural destruction and a mixed infiltrate of cells, including B cells at different stages of maturation, such as immunoblasts, small lymphoid cells, and plasmacytoid cells/plasma cells. Necrosis is also identified in the left upper area (black arrow; Hematoxy–eosin stain 200×). (**B**) Immunoblasts showed Hodgkin/Reed Sternberg (HRS)-like morphology patterns (black arrow; Hematoxy–eosin stain 400×). (**C**) Epstein–Barr encoding RNA (EBER) in situ hybridization positive in cells of various sizes including Hodgkin/Reed–Sternberg (HRS)-like immunoblasts (black arrow; in situ hybridization/immunohistochemistry stain 200×).

**Figure 3 diagnostics-15-01517-f003:**
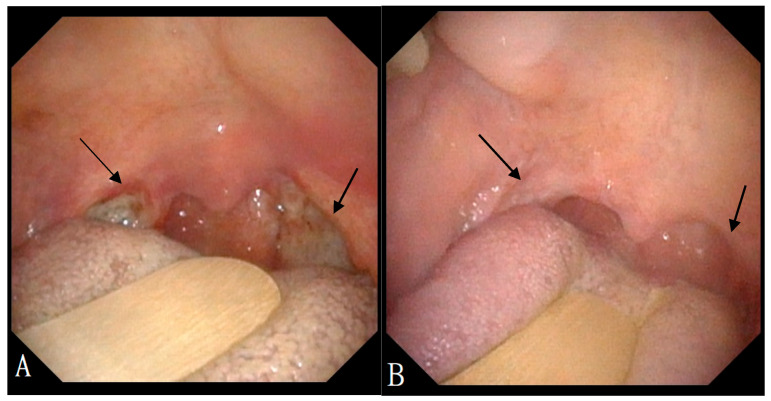
(**A**) Physical findings of the oropharynx showing ulceration and necrotic tissue in bilateral tonsillar fossa prior to methotrexate discontinuation (black arrows). (**B**) Resolution of tonsillar ulceration and normalization of the oropharyngeal mucosa following the cessation of methotrexate therapy (black arrows).

**Table 1 diagnostics-15-01517-t001:** Summary of published cases of methotrexate-associated lymphoproliferative disorders (MTX-LPDs).

Study	Patient Population	Location	EBV Status	Histology	Treatment	Outcome
Dojcinov et al. (2010) [2]	26 patients on immunosuppressants	Oral mucosa, skin, GI tract	EBV+	EBV-positive mucocutaneous ulcer (EBV-MCU)	MTX withdrawal ± rituximab	Majority showed spontaneous regression
Kikuchi et al. (2010) [3]	1 RA patient on MTX	Oral cavity (palate)	EBV+	Hodgkin-like LPD	MTX cessation	Complete remission
Yamaguchi et al. (2024) [7]	1 RA patient on MTX	Sacrum (bone)	Not specified	Polymorphic B-cell LPD	MTX cessation	Tumor regressed spontaneously
Satou et al. (2021) [5]	12 patients with skin lesions	Skin	EBV−	B-cell LPD	MTX cessation ± chemo	Most had a favorable outcome
Shiraiwa et al. (2020) [4]	21 patients with mucosal lesions	Oropharynx, GI tract	EBV+	EBV-MCU/polymorphic LPD	MTX cessation ± local therapy	85% showed spontaneous remission
Salloum et al. (1996) [10]	RA and rheumatic patients on MTX	Lymph nodes, GI tract, oropharynx	Mixed	Polymorphic or Hodgkin-like LPD	MTX cessation	Multiple cases regressed
Present Case (2025)	70-year-old woman with RA	Bilateral tonsils, mediastinum, axilla	EBV+	Polymorphic LPD with Hodgkin-like features	MTX cessation	Complete resolution

## Data Availability

All data used are available within this article.

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
