# Peer review of "Protracted Tonsillitis as an Atypical Initial Manifestation of Methotrexate-Induced EBV-Positive Lymphoproliferative Disorder in Rheumatoid Arthritis: A Case Report and Literature Review"

_diagnostics, 2025, doi:10.3390/diagnostics15121517_

Round 1
Reviewer 1 Report
Comments and Suggestions for Authors
Dear Authors,
Thank you for the opportunity to review your manuscript. Your case report presents an extremely rare initial manifestation of methotrexate (MTX)-associated lymphoproliferative disorder (LPD) with bilateral necrotic tonsillitis, offering valuable insights into this condition. Your work highlights an important diagnostic challenge in immunosuppressed patients and underscores the necessity of considering rare complications like LPD in atypical clinical scenarios. The observed spontaneous regression upon MTX withdrawal has significant clinical implications, emphasizing the potential for conservative management rather than unnecessary chemotherapy. Overall, the manuscript is well-structured, clearly written, and flows logically from case presentation to discussion and literature review. The atypical presentation and successful conservative management offer valuable lessons. Your manuscript is a meaningful contribution to the field and reinforces the importance of clinical vigilance when evaluating immunosuppressed patients with unusual symptoms.
Minor Suggestions for Improvement:
-
The paper references Epstein-Barr virus-encoded RNA (EBER) in situ hybridization as essential for diagnostic clarification. It may be beneficial to briefly define "EBER" upon its first mention for readers unfamiliar with the acronym.
-
Please add arrows to Figures 1–3 to highlight key findings for improved readability.
Your manuscript is well-executed and provides valuable insights into a rare but significant complication of MTX therapy. I recommend acceptance after minor revisions.
Thank you for your thoughtful contributions to this important topic.
Sincerely,
Reviewer 2 Report
Comments and Suggestions for Authors
Due to CT scan also revealed multiple necrotic lymphadenopathies in the cervical region, did authors ever consider about neck lymph node core needle biopsy for diagnosis in the beginning, instead of tonsillectomy? Also, is there any other diagnostic options could be suggested instead of tonsillectomy like neck lymph nodes excisional biopsy, core needle biopsy... etc. for EBV-positive LPD? It could be mentioned in the Discussion.
Author Response
Comment 1: Due to CT scan also revealed multiple necrotic lymphadenopathies in the cervical region, did authors ever consider about neck lymph node core needle biopsy for diagnosis in the beginning, instead of tonsillectomy? Also, is there any other diagnostic options could be suggested instead of tonsillectomy like neck lymph nodes excisional biopsy, core needle biopsy... etc. for EBV-positive LPD? It could be mentioned in the Discussion.
Response 1: We thank the reviewer for this insightful comment. We have now addressed this point in the Discussion section. While tonsillectomy was pursued primarily due to the patient’s localized symptoms and prominent tonsillar findings, we agree that less invasive diagnostic modalities, such as image-guided core needle biopsy (CNB) or excisional lymph node biopsy, represent viable alternatives for establishing a diagnosis in cases of suspected EBV-positive LPD. In the revised manuscript, we discuss the utility of CNB in this context and reference recent evidence supporting its diagnostic accuracy in LPDs (Ferrari et al., 2025). This addition strengthens the clinical relevance of the case and emphasizes individualized diagnostic planning based on disease distribution, tissue accessibility, and procedural yield.
Reviewer 3 Report
Comments and Suggestions for Authors
A good manuscript on MTX induced EBV LPD tonsillitis. However, minor revision is required, with regards to labelling of all the figures provided.
Additionally, the oropharynx and tonsils photos need to be replaced with a better one. Both tonsils are hardly visible!
Thank you.

Author Response
We thank the reviewer for the encouraging feedback and helpful suggestions. In response to your comment:
Comment 1: A good manuscript on MTX induced EBV LPD tonsillitis. However, minor revision is required, with regards to labelling of all the figures provided.
Response 1: We have now revised all figure panels and added clear arrow annotations to highlight relevant findings, including anatomical landmarks and pathologic features, for improved interpretability.
Comment 2: Additionally, the oropharynx and tonsils photos need to be replaced with a better one. Both tonsils are hardly visible!
Response 2: We acknowledge the suboptimal clarity of the oropharyngeal images. Unfortunately, these are the only available clinical photographs in the electronic medical record, and no better-quality images are retrievable retrospectively. Nonetheless, we have confirmed that the existing images correspond to the time of initial presentation and have enhanced their labeling to aid visual interpretation.
We would like to thank the reviewer for the insightful and constructive comments.